# Bioinspired enzymatic compartments constructed by spatiotemporally confined in situ self-assembly of catalytic peptide

Yaling Wang [1,5], Tiezheng Pan [2,5], Xuewen Wei[1], Fangcui Su[3], Ang Li[1], Yifan Tai[1], Tingting Wei[1], Qian Zhang[1], Deling Kong [1,4✉] & Chunqiu Zhang [1✉]

Enzymatic compartments, inspired by cell compartmentalization, which bring enzymes and substrates together in confined environments, are of particular interest in ensuring the enhanced catalytic efficiency and increased lifetime of encapsulated enzymes. Herein, we constructed bioinspired enzymatic compartments (TPE-Q18H@GPs) with semi-permeability by spatiotemporally controllable self-assembly of catalytic peptide TPE-Q18H in hollow porous glucan particles (GPs), allowing substrates and products to pass in/out freely, while enzymatic aggregations were retained. Due to the enrichment of substrates and synergistic effect of catalytic nanofibers formed in the confined environment, the enzymatic compartments exhibited stronger substrate binding affinity and over two-fold enhancement of second-order kinetic constant ($k_{cat}/K_m$) compared to TPE-Q18H nanofibers in disperse system. Moreover, GPs enabled the compartments sufficient stability against perturbation conditions, such as high temperature and degradation. This work opens an intriguing avenue to construct enzymatic compartments using porous biomass materials and has fundamental implications for constructing artificial organelles and even artificial cells.

[1] State Key Laboratory of Medicinal Chemical Biology, Key Laboratory of Bioactive Materials of Ministry of Education and College of Life Sciences, Nankai University, Tianjin, China. [2] School of Life Science and Technology, Tokyo Institute of Technology, Yokohama, Japan. [3] Key Laboratory for Molecular Enzymology and Engineering of Ministry of Education, School of Life Sciences, Jilin University, Changchun, China. [4] Frontiers Science Center for Cell Responses, Nankai University, Tianjin, China. [5] These authors contributed equally: Yaling Wang, Tiezheng Pan. ✉email: kongdeling@nankai.edu.cn; zhangcq@nankai.edu.cn

In nature, the high efficiency of biochemical reaction networks requires cells to bring appropriate enzymes and substrates together in spatiotemporally confined compartments, serving to maintain the metabolic order, concentrate reactants to drive unfavorable reactions, or prevent enzymes and unstable intermediates from harmful cellular conditions[1]. The enzymatic compartment can be regarded as a highly structured matrix wrapped in a semi-permeable boundary, which is evolved to stabilize and protect catalytic active species by separating the reaction space from the surrounding environment (e.g., protease and toxic chemicals), ensuring that the reactions could proceed stably and controllably[2]. Such appealing features not only provide spatial-temporal benefits in organisms but also inspire us to construct enzymatic compartments, translating biochemical pathways into robust and controllable industrial processes.

Conceptually, significant efforts have been made to mimic enzymatic compartments by employing vesicles[3–5], polymersomes[6–9], protein-cages[10–13], and reverse micelle droplets[14–17] as the outer boundary materials to physically entrap natural enzymes. To render the semi-permeability of boundary and enable the exchange of substrates and products with surroundings, various strategies have been developed, such as making porous polymersomes[7,18], introducing pore proteins or bio-pores into the vesicle membrane[19,20], and disturbing the stability of membrane by using an external stimulus[21]. All these considerations concerned how external substrates could reach the interior of the compartments, which required the precise design of the structures of boundary and substrates. Alternatively, one approach to tackle this challenge was to put the enzymes and all reacting molecules into compartments from the outset[22]. However, the enzymatic reactions were expected to occur from the beginning of preparing the systems without the spatiotemporal control.

Natural porous structures from biomass are of interest in de novo fabrication of diverse functional materials, especially for energy conversion and storage[23,24]. Thus, nature probably can provide us the best biomaterials for the construction of enzymatic compartments. The branched β-(1,3; 1,6)-glucan particles (GPs) from yeast cell walls are porous hollow microspheres with physico-chemically stable and biocompatible properties[25]. The permeability of the GPs allows small molecules and biomolecules (e.g., Rifampicin, RNA, DNA) to enter and leave the capsule interior freely[25–27]. Apart from the permeability, this nontoxic capsule not only provides a cavity for loading cargoes because of the hollow cavity inside the GPs together with a rather "biofriendly" package mode but also potentially contributes a confined reaction space just as their inherently protective and filtered responsibility in living systems, which makes GPs a kind of promising candidate for constructing enzymatic compartments in non-living systems.

Herein, we presented the first example to engineer bioinspired enzymatic compartments by in situ incorporation and self-assembly of catalytic peptides within the interior of GPs. Beyond the spherical structure of proteins, self-assembly peptides could form variable morphologies, including 1D nanofibers[28–31], nanotubes[32–34], 2D nanosheets[35,36], and even crystal structure[37–39], and these micro-scale structures would be retained in the cavity of GPs to overcome the limitation of the permeability of GPs (i.e., the biomolecular leakage after the encapsulation). Meanwhile, pH[40–43], salt[44,45], and temperature[46–48] responsive behaviors of peptide self-assemblies enable them to achieve spatiotemporally controllable assembly in a confined environment. Moreover, recently self-assembly peptides with enzymatic properties have been well studied[49–53], which makes them promising candidates to construct artificial enzymatic compartments instead of natural enzymes. Our starting point is based on the family of salt-responsive self-assembly peptide Q11 (Ac-QQKFQFQFEQQ-Am) developed by Collier and Messersmith[54–57] that have shown the ability to self-assemble into well-defined nanofibrils triggered by regulating ionic strength. Of the available self-assembly peptides, in our previous work[58], the Q11 systems have been well expanded as the catalytic platform, providing a suitable model as the catalytic part to construct enzymatic compartments.

Our designed catalytic peptide is a modification of Q11 named TPE-Q18H (Fig. 1), which is composed of three segments with different functions: (i) the N-terminal tetraphenylethylene (TPE) was chosen as the self-assembly indicator due to its aggregation-induced emission (AIE) property[59]; (ii) the C-terminal histidine (H) residue was relevant to the hydrolysis of activated esters, a typical model reaction for mimicking hydrolase[60–62]; (iii) the salt-responsive peptide Q11 modified with RKRK for increasing the water solubility of TPE and GS as a spacer to make catalytic residue H more flexible. Here, peptide TPE-Q18H was incorporated and in situ self-assembled in the cavity of GPs by regulating the ionic strength of the system to construct enzymatic compartments (TPE-Q18H@GPs). As with prior studies[50,58], we used 4-nitrophenyl acetate (pNPA) as the substrate to monitor the

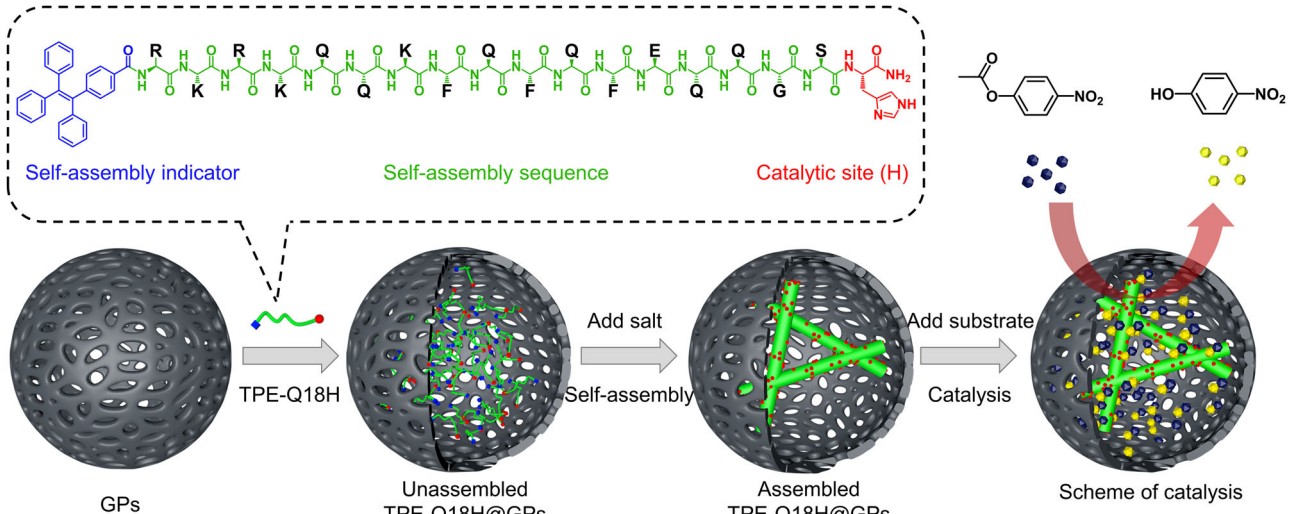

**Fig. 1 Illustration of temporal and spatial controlled GPs-based enzymatic compartments.** The catalytic peptide (TPE-Q18H) was assembled in GPs by regulating the ionic strength and promoting the ester hydrolysis reaction with a higher catalytic efficiency.

catalytic performance (Fig. 1). We first confirm the spatio-temporally controllable self-assembly of TPE-Q18H in GPs and then demonstrate the semi-permeable ability of the system, retaining TPE-Q18H fibrillar structures and allowing small molecules to enter and leave freely. Furthermore, we measure the catalytic activities and kinetic constants of TPE-Q18H@GPs and TPE-Q18H nanofibers, respectively, to confirm the essential role of the confined space in GPs. Finally, the thermostability and resistance to hydrolysis of TPE-Q18H@GPs and TPE-Q18H nanofibers were also investigated. All these demonstrate that the confined self-assembly makes TPE-Q18H@GPs exhibit higher catalytic efficiency and GPs provide sufficient protective effect towards high temperature and degradation.

## Results and discussion

**Materials preparation and characterizations.** The peptide TPE-Q18H was synthesized by solid-phase peptide synthesis (SPPS), purified by reverse phase HPLC, and the purity and molecular weight were confirmed by HPLC and MALDI-TOF MS (Supplementary Fig. 1 and Supplementary Fig. 2). Transmission electron microscopy (TEM) and atomic force microscopy (AFM) were used to obtain the morphology of the self-assembled TPE-Q18H nanofibrillar structures (Fig. 2a, b) with widths of about 7 nanometers and relatively homogeneous featured heights of 6 nanometers (Supplementary Fig. 3) (Experimental details are given in Supplementary Methods). The addition of other moieties onto Q11 terminus did not influence the formation of fibrils, consistent with previous reports[55]. Besides, we characterized the conformation of TPE-Q18H through fluorescence spectro-photometry (Experimental detail is given in Supplementary Methods). TPE-Q18H nanofibers in PBS exhibited dramatically enhanced fluorescent intensity at 466 nm of TPE compared to the peptide monomer state in water (Fig. 2g), meaning that peptide TPE-Q18H formed aggregations. TPE-Q18H nanofibers also exhibited pronounced enhancement at 488 nm maxima of thio-flavin T (Th T) fluorescent spectrum (Fig. 2h), a typical β-sheet

indicator[63], providing strong evidence of the formation of β-sheet structures. Then, the timeframe of the salt-responsive assembly process of TPE-Q18H was further investigated by monitoring the fluorescence of Th T. After adding salt to improve the ionic strength of TPE-Q18H solution, we found that the fluorescence intensity of Th T increased immediately and tended to be stable after $t = 30$ min, indicating that the assembly reached equilibrium at this time point (Supplementary Fig. 4). The GPs were prepared from Baker's yeast with repeated alkaline-and-acid and solvent treatments[64]. Empty GPs are porous, about 3–5 micrometers in diameter under the scanning electron microscope (SEM) (Fig. 2c, Supplementary Fig. 5) and translation electron microscope (Fig. 2e) (Experimental details are given in Supplementary Methods). The surface adsorption properties of GPs were mea-sured to investigate the pore size (Experimental detail is given in Supplementary Methods). The results showed that the BET surface area and average pore diameter of GPs were 12.84 m²/g and 4.14 nm, respectively. The adsorption-desorption curve of GPs is a type II isotherm (Fig. 2d), which may occur in the process of reversible adsorption of single multi-layer on meso-porosity solids.

**Exploration of enzymatic compartments fabrication strategy.** TPE-Q18H@GPs were fabricated in a spatiotemporally con-trollable manner: first, peptide TPE-Q18H was dissolved in solution (acetonitrile/water = 3:7) to get a clear solution and the peptide was in the monomer state; then, the GPs powder was added into the TPE-Q18H solution and the peptide was inter-nalized into GPs by capillarity owing to the porous structure of GPs; finally, 1 × PBS buffer was added to trigger the self-assembly of TPE-Q18H and the TPE-Q18H@GPs was obtained by cen-trifugation and washed with 1 × PBS for three times. The loading capacity ($W_{TPE-Q18H}/W_{GPs}$) was 90 μg/mg and the encapsulation efficiency was over 90% (Supplementary Fig. 7), conducted by UV–vis measurements (Supplementary Fig. 6). Compared with empty GPs, TPE-Q18H@GPs showed corrugation by TEM

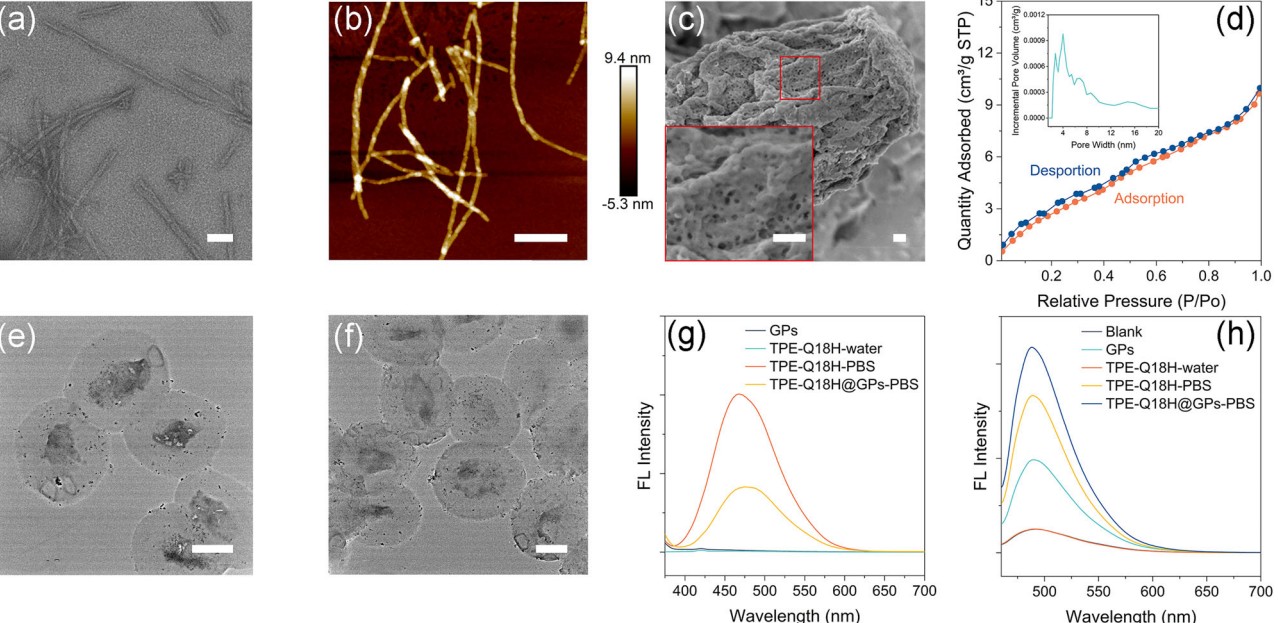

**Fig. 2 Characterization of TPE-Q18H, GPs, and TPE-Q18H@GPs. a** TEM image of TPE-Q18H nanofibers. Scale bar: 50 nm. **b** AFM image of TPE-Q18H nanofibers. Scale bar: 200 nm. **c** SEM image and locally enlarged view of GPs. Scale bar: 100 nm. **d** Nitrogen adsorption isotherms and pore size distribution (inset) calculated based on DFT method for GPs. **e** TEM images of GPs and **f** TEM images of TPE-Q18H@GPs. Scale bar: 2 μm. **g** Fluorescence spectra of TPE-Q18H, GPs, and TPE-Q18H@GPs. $\lambda_{ex} = 365$ nm. **h** Fluorescence spectra of thioflavin T in presence of TPE-Q18H, GPs, and TPE-Q18H@GPs. $\lambda_{ex} = 440$ nm. TPE-Q18H: 90 μg/mL; GPs: 1 mg/mL; TPE-Q18H@GPs: 90 μg/mg/mL; Th T: 100 μM.

measurement (Fig. 2f), suggesting TPE-Q18H was successfully encapsulated into GPs. To further verify the assembly of TPE-Q18H in GPs, the fluorescence intensity changes of TPE and Th T were investigated (Experimental detail is given in Supplementary Methods). From Fig. 2g, h, we could see that TPE-Q18H@GPs in PBS exhibited dramatically enhanced fluorescent intensity at 466 and 488 nm, meaning that peptide TPE-Q18H formed β-sheet structures in GPs. In the meanwhile, we observed the fluorescence of Th T reached the highest level immediately after adding salt in TPE-Q18H@GPs (Supplementary Fig. 8), demonstrating that peptide TPE-Q18H assembled completely in GPs within one minute. We hypothesize that the ultra-fast assembly process of TPE-Q18H@GPs is because the peptides are already highly concentrated in GPs and the very close distance accelerates the process of nucleation and recruitment of free peptide molecules.

Furthermore, empty GPs were labeled by 5-DTAF(5-(4,6-Dichlorotriazinyl) Aminofluorescein) and then TPE-Q18H self-assembled in the DTAF-labeled GPs. Confocal laser scanning microscopy (CLSM) was used to investigate the co-localization of TPE-Q18H and GPs (Experimental detail is given in Supplementary Methods). As shown in Fig. 3, the blue fluorescence of TPE-Q18H was enclosed within the green fluorescence from DTAF, indicating the successful encapsulation of TPE-Q18H in GPs. In order to further prove the conformation of TPE-Q18H in GPs, the Fourier Transform infrared spectrometer (FTIR) was investigated. The FTIR spectra of assembled TPE-Q18H, empty GPs and TPE-Q18H @GPs were obtained by KBr tablet method. Absorbance spectrum of assembled TPE-Q18H and TPE-Q18H @GPs shared two peaks in amide I region at 1631 and 1674 cm$^{-1}$ (Supplementary Fig. 9), indicating antiparallel β-sheet structures, which was in agreement with Th T results. These results were in direct agreement with the original peptide Q11 and confirmed that the confined space did not alter the self-assembly conformation of TPE-Q18H.

**Semi-permeability of constructed compartments.** An attractive feature of our design was to create a semi-permeable compartment where the substrates/products could enter/leave the compartment without hindrance and the enzymatic peptide aggregations were retained. Here, the semi-permeability was assessed by monitoring the release of TPE-Q18H and Rhodamine from GPs. Rhodamine is a traditional fluorescent dye and here it is used as the representative of substrates/products. First, the Rhodamine-loaded GPs (Rhodamine@GPs) were prepared by simply pretreating empty GPs with Rhodamine. After centrifuging and resuspending three times, CLSM and fluorescence spectrophotometer were used to detect the fluorescent intensity changes of Rhodamine during this process. Before the centrifugation, the strong red fluorescence of Rhodamine was observed in the interior of GPs (Fig. 4a, c), indicating that Rhodamine could enter GPs freely. After three times centrifugation, the remaining fluorescence of Rhodamine in GPs was hardly detected (Fig. 4b, c), but the supernatant solution showed strong fluorescence (Supplementary Fig. 10). This data illustrated small molecule Rhodamine was free to pass in and out, and the releasing kinetics of Rhodamine from GPs showing that 50% Rhodamine was released in the first centrifugation, and all Rhodamine was released after 5 centrifugations (Supplementary Fig. 11). In contrast, TPE-Q18H@GPs solution was performed the same operation as Rhodamine@GPs. We found that the fluorescence intensity of TPE from the self-assembly just changed within a very narrow window (Fig. 4d–f) and almost no fluorescence was recorded in the supernatant solution (Supplementary Fig. 12), indicating that the TPE-Q18H nanofibers could be retained within GPs. As expected, even in the TPE-Q18H@GPs system, Rhodamine was still able to permit in and out freely by monitoring its fluorescence change (Supplementary Figs. 13 and 14).

**Catalysis of enzymatic compartments.** Having demonstrated the semi-permeability of this system, a hydrolysis model was then applied to investigate the confined effect on reaction kinetics. The hydrolytic activities of TPE-Q18H nanofibers and TPE-Q18H@GPs were measured using 4-nitrophenyl acetate (pNPA) as substrate, which is a widely assayed and standardized chromogenic substrate by monitoring the appearance of the yellow hydrolyzed product, 4-nitrophemolate (pNP). Thus, checking the release of 4-nitrophenol from GPs could also confirm the permeability and separative capacity of our system. After adding the substrate to the TPE-Q18H@GPs solution, 4-nitrophenol was generated within the enzymatic compartments, but then diffused into the bulk solution. Visible imaging in situ revealed that the visible yellow color came out just in minutes after the addition of pNPA to the reaction system. After centrifugation, the product was almost totally in the supernatant (Fig. 5a and Supplementary Fig. 15). This result meant that the products could be separated by a very simple centrifugal operation, which allowed for a facile collection of enzymatic compartments from the reaction system.

The catalytic activity was measured by monitoring the concentration of the hydrolyzed product pNP as a function of time. We found that GPs were proved to have no contribution to the catalytic activity and TPE-Q18H@GPs exhibited dramatic hydrolytic activity towards pNPA (Supplementary Fig. 16). In addition, the activities of TPE-Q18H and TPE-Q18H@GPs along with the assembly process were measured. The hydrolysis rates of pNPA increased rapidly in both TPE-Q18H (Supplementary Fig. 17) and TPE-Q18H@GPs (Supplementary Fig. 18) groups after adding salt, demonstrating that the catalytic performance emerged as long as the assembly started.

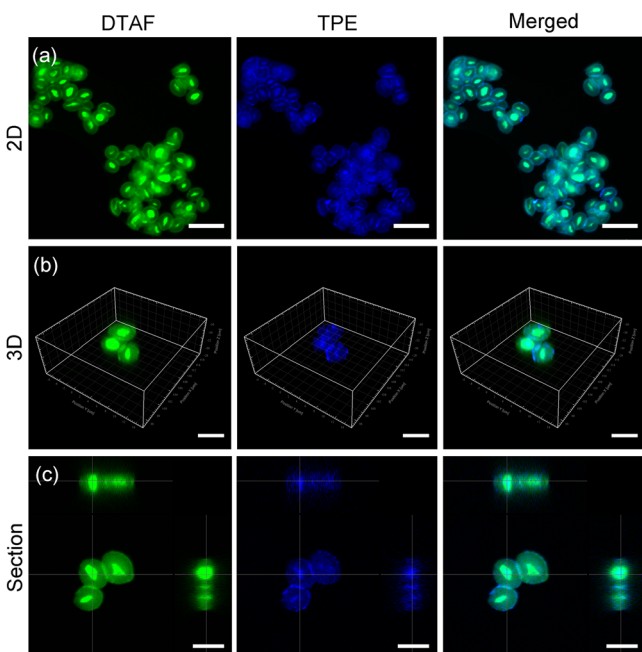

**Fig. 3 Confocal laser scanning microscope images of TPE-Q18H@GPs.** **a** 2D images of TPE-Q18H@GPs. Scale bar: 10 μm. **b** 3D images of TPE-Q18H@GPs. Scale bar: 5 μm. **c** One section of 3D images of TPE-Q18H@GPs. Scale bar: 5 μm. GPs were labeled by DTAF. The 3D and section images were obtained by scanned and photographed along the Z-axis in CLSM, and reconstructed by Imaris software. $\lambda_{ex} = 405$ nm for TPE and $\lambda_{ex} = 488$ nm for DTAF.

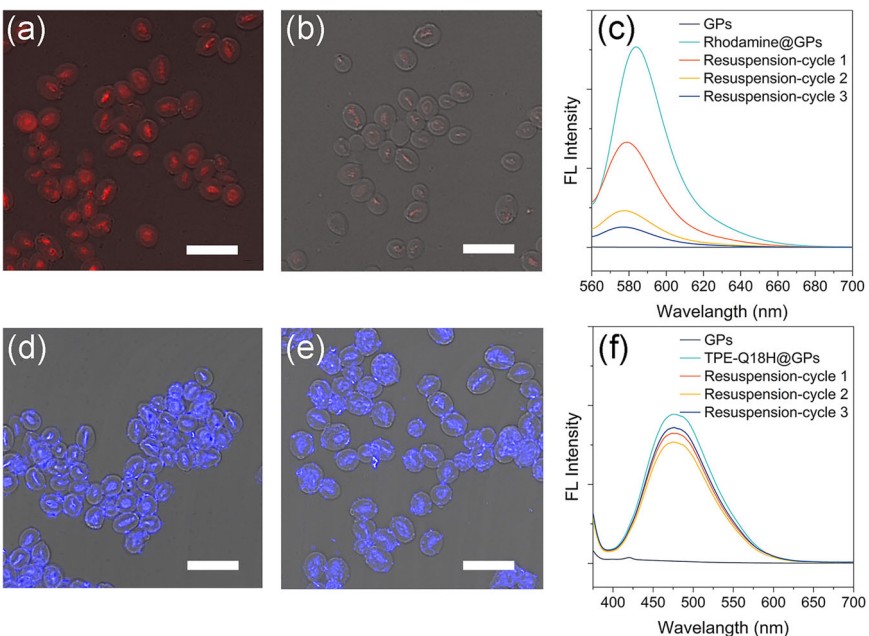

**Fig. 4 Semi-permeability was assessed by CLSM and fluorescence spectrophotometry.** CLSM images of **a** free GPs were incubated with Rhodamine for 10 min (Rhodamine@GPs) and **b** Rhodamine@GPs after three times centrifugation. **c** Fluorescence intensity changes of Rhodamine@GPs during three times centrifugation. CLSM images of **d** TPE-Q18H@GPs and **e** TPE-Q18H@GPs after three times centrifugation. **f** Fluorescence intensity changes of TPE-Q18H@GPs during three times centrifugation. Scale bar: 10 μm. In fluorescence spectra, $\lambda_{ex} = 365$ nm for TPE and $\lambda_{ex} = 540$ nm for Rhodamine, in CLSM images $\lambda_{ex} = 405$ nm for TPE and $\lambda_{ex} = 561$ nm for Rhodamine.

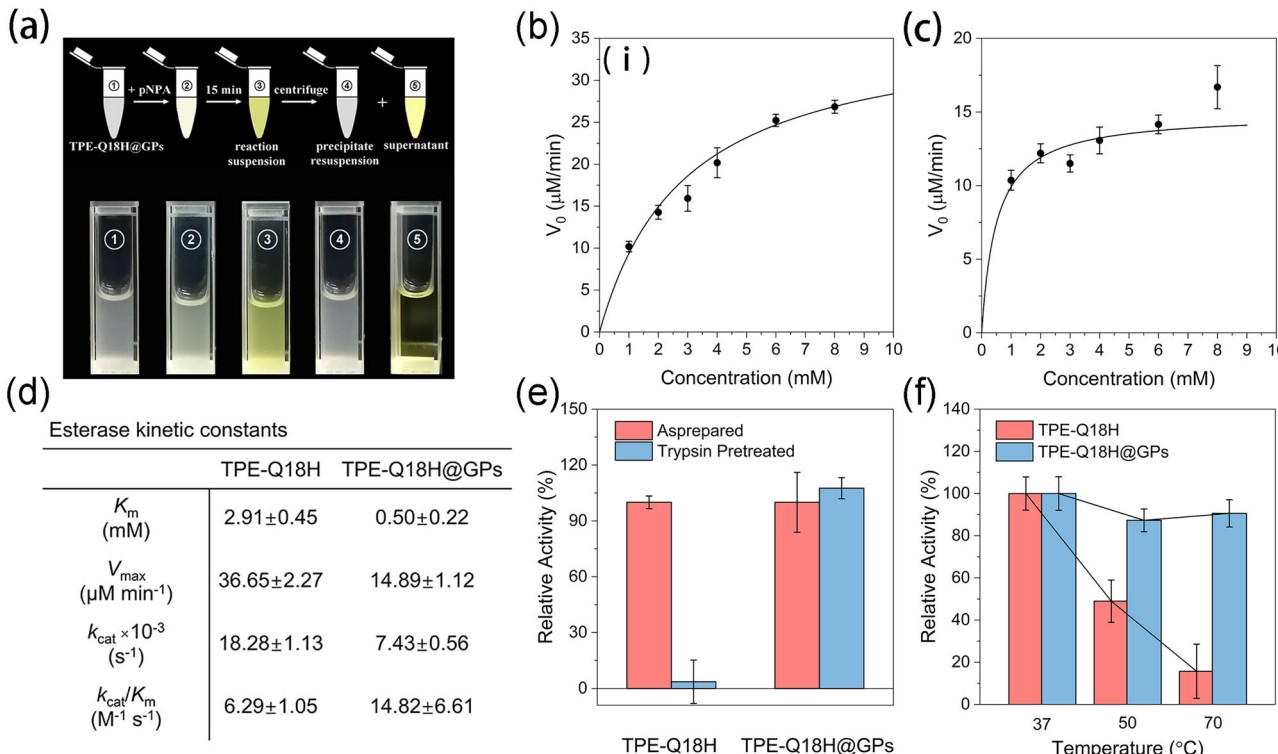

| Esterase kinetic constants | | |
|---|---|---|
| | TPE-Q18H | TPE-Q18H@GPs |
| $K_m$ (mM) | 2.91±0.45 | 0.50±0.22 |
| $V_{max}$ (μM min⁻¹) | 36.65±2.27 | 14.89±1.12 |
| $k_{cat} \times 10^{-3}$ (s⁻¹) | 18.28±1.13 | 7.43±0.56 |
| $k_{cat}/K_m$ (M⁻¹ s⁻¹) | 6.29±1.05 | 14.82±6.61 |

**Fig. 5 The ester hydrolysis activity of TPE-Q18H nanofibers and TPE-Q18H@GPs. a** The reaction and separation procedure of product 4-nitrophenol with compartments. Non-linear fitting of pNPA hydrolysis catalyzed by **b** TPE-Q18H nanofibers and **c** TPE-Q18H@GPs. **d** Esterase kinetic constants of TPE-Q18H nanofibers and TPE-Q18H@GPs. **e** Catalytic activities of TPE-Q18H nanofibers and TPE-Q18H@GPs after the treatment of trypsin. **f** Temperature-dependent activities of TPE-Q18H nanofibers and TPE-Q18H@GPs. Data are presented as the mean ± SD (dot with error bar) of $n = 3$ independent measurements.

Michaelis–Menten kinetics is a well-known model of enzyme kinetics that distinguishes enzymes from other chemical catalysts. To determine whether TPE-Q18H and TPE-Q18H@GPs followed characteristic enzyme kinetics, the initial catalytic hydrolytic rates of pNPA were measured as a function of various substrate concentrations. The Michaelis–Menten model fitting plots of TPE-Q18H and TPE-Q18H@GPs were obtained in Fig. 5b, c. As shown in Fig. 5d, TPE-Q18H@GPs revealed over two-fold enhancement of the second-order kinetic constant ($k_{cat}/K_m$ = 14.82 ± 6.61 M$^{-1}$ s$^{-1}$, where $k_{cat}$ is the catalytic rate constant and $K_m$ is the apparent Michaelis constant) compared to that of TPE-Q18H nanofibers ($k_{cat}/K_m$ = 6.29 ± 1.05 M$^{-1}$ s$^{-1}$), although TPE-Q18H nanofibers exhibited a higher catalytic constant ($k_{cat}$ = 18.28 ± 1.13 × 10$^{-3}$ M$^{-1}$ s$^{-1}$) than TPE-Q18H@GPs ($k_{cat}$ = 7.43 ± 0.56 × 10$^{-3}$ M$^{-1}$ s$^{-1}$). Such an increase of the second-order kinetic constant for enzymatic compartments was from the dramatic reduction of the apparent Michaelis constant $K_m$ (0.5 ± 0.22 mM), which was supposed to be due to the structural advantages of compartments. We hypothesized that one reason was that the porous structure enriched the substrates and increased the substrate concentration around catalytic sites; the other was that the confined space shortened the distance of the interior TPE-Q18H nanofibers and forced them to form more complicated and three-dimensional catalytic microenvironment, which resulted in the enhancement of substrate-binding affinity, while TPE-Q18H nanofibers in solution were lack of such synergistic effect.

## Protection effect of GPs-based compartments

To reveal the protective effect of GPs on our system, the catalytic activities of TPE-Q18H@GPs and TPE-Q18H nanofibers were examined after the treatment of protease and high temperature. On account of the hydrolytic ability of trypsin towards pNPA (Supplementary Fig. 19), the activities of TPE-Q18H nanofibers with trypsin were calculated by deducting the activity of trypsin, and activities of TPE-Q18H@GPs with trypsin were calculated by deducting the activity of trypsin with GPs. After incubating with trypsin (0.1 mg/mL) for 5 min, the TPE-Q18H@GPs retained ~100% of its activities (Fig. 5e). Meanwhile, the activity of TPE-Q18H nanofibers only remained 3.6% left (Fig. 5e). Then the thermostability of TPE-Q18H@GPs and free TPE-Q18H nanofibers were investigated. The incubation of TPE-Q18H nanofibers above 50 °C resulted in an obvious decline of activities (up to 85% at 70 °C) (Fig. 5f). On the contrary, TPE-Q18H@GPs kept 90% relative activities even at 70 °C (Fig. 5f). All these results demonstrated that GPs could significantly provide protection and maintain the stability of the catalytic environment.

## Conclusion

In this study, we have successfully developed a facile approach to construct enzymatic compartments by in situ self-assembly of catalytic peptide TPE-Q18H in GPs. The self-assembly of peptides is controlled by salt, and the encapsulation follows the self-assembly "trigger", making the whole fabrication procedure spatiotemporally controllable. The naturally porous structure of GPs allows the substrates and products to pass in/out freely, while the self-assembly aggregations retain the catalytic peptides in the interior of the cavity, making the compartments semi-permeable in the reaction system. In accordance with other histidine-containing self-assembly peptides[49,50,52], TPE-Q18H@GPs also exhibit hydrolytic capacity and show a more than two-fold value of second-order kinetic constant compared to TPE-Q18H nanofibers, supposed to be due to the enrichment of substrates and synergistic effect of catalytic nanofibers in the confined space. Finally, we also demonstrate TPE-Q18H@GPs with dramatic ability against perturbation conditions including high temperatures and protease digestion. Such diverse advantages inform us of the possibilities to create enzymatic compartments by using porous biomass materials and expand approaches in the construction of artificial organelles and even artificial cells.

## Methods

**Materials**. Fmoc-amino acids, diisopropylethylamine (DIPEA), CLEAR-Amide Resin, and O-(7-azabenzotriazole-1-yl)-1,1,3,3-tetramethyluronium hexafluorophosphate (HATU) were purchased from BO MAI JIE Technology Co., Ltd. (Beijing, China). Saccharomyces cerevisiae baker's yeast was obtained from Anqi yeast Co., Ltd. Nitrophenyl acetate (p-NPA), thioflavin T (ThT), and Trypsin was purchased from Sigma-Aldrich. TPE was purchased from Beijing Hwrkchemical Co,. Ltd and other chemicals and solvents were commercially purchased.

**Peptide synthesis**. Peptide TPE-Q18H was synthesized by standard SPPS on CLEAR Amide Resin. Fmoc-protected group was removed by 20% piperidine in anhydrous N, N'- dimethylformamide (DMF), and HATU and DIPEA were chosen as the coupling reagents. In the last coupling step, carboxylated tetraphenylethylene (TPE-COOH) was used to cap the amine group of the peptide. The peptide derivatives were cleaved from the resin by a mixture of TFA, Milli-Q water, and Triisopropylsilane for 3 h at room temperature. The mixture was then evaporated and precipitated by ice-cold diethyl ether. The resulting precipitate was washed by ice-cold diethyl ether and dissolved in DMSO for HPLC purification. The purified and molecular weight of the product was characterized by HPLC and MALDI-TOF MS.

**Preparation of glucan particles (GPs)**. Glucan particles were obtained from Saccharomyces cerevisiae baker's yeast (Angel, Hubei, China) according to the alkaline-and-acid method[25]. Briefly, the yeast (10 g) was suspended in pre-heated 1.0 M NaOH (freshly prepared and heated to 80 °C) and stirred for 1 h. Then it was centrifuged at 8000 × $g$ for 10 min. The precipitate was then resuspended in 100 mL Milli-Q water and washed three times. The pH was adjusted to 5 and incubated for 1 h at 55 °C. Then it was centrifuged, and the precipitate was washed twice with 100 mL Milli-Q water, four times with 100 mL isopropanol, and twice with 100 mL acetone. The precipitate was dried at room temperature to produce glucan particles powder and stored at −20 °C.

To prepare the DTAF-labeled GPs, 10 mg GPs were suspended in 1 mL 0.1 M Na$_2$CO$_3$ buffer (pH 9.2). By adding 250 μL 5-DTAF (DMSO,1 mg/mL), the mixture was stirred in dark at room temperature. After 16 h, 2 mM Tris HCl buffer (pH 6.8) was added and incubated for an extra 15 min. Finally, it was centrifuged (8000 rpm/min) for 8 min and washed three times with 2 mM Tris HCl buffer, three times with Milli-Q water and twice with acetone. The precipitate was dried at room temperature to produce DTAF-labelled GPs and stored at −20 °C.

**Preparation of TPE-Q18H nanofibers and TPE-Q18H@GPs compartment**.
Peptide TPE-Q18H powder was dissolved in the mixture of 30% acetonitrile and 70% Milli-Q water at a concentration of 15 mg/mL, then the peptide solution was diluted with 1 × PBS buffer (pH 7.4) and the solution of nanofibers (33.42 μM) was obtained after storing for 24 h at room temperature.

For the preparation of TPE-Q18H@GPs, 30 μL TPE-Q18H solution (15 mg/mL) and 5 mg dry particles were mixed forming a thick paste, followed by centrifuging tube briefly (~5 s) to bring the mixture to the bottom of the tube. Repeating the mixing and centrifugation obtained a homogenous paste. After 2 h, the paste was lyophilized to remove the water. The lyophilized powder was resuspended with 1 mL 1 × PBS to obtain a good dispersion TPE-Q18H@GPs particles solution and stocked at 4 °C. When measuring the enzyme activities, add another 4 mL 1 × PBS to get a diluted solution (TPE-Q18H: 33.42 μM; GPs: 1 mg/mL).

The loading capacity was determined by UV–vis spectrophotometry (HITACHI U-3900). The UV–vis spectra of TPE-Q18H 1 × PBS solution in different concentrations were obtained on a UV–vis spectrophotometer from 190 to 450 nm to make a standard curve. Three successive wavelength scans were taken to average for each concentration. TPE-Q18H@GPs was centrifuged at 8000 rpm/min for 5 min. Supernatants were analyzed for TPE-Q18H content using UV–vis spectroscopy with the help of a standard plot. Three successive wavelength scans were taken to average. The encapsulation efficiency (EE %) and the loading capacity (LC, μg mg$^{-1}$) were calculated as follows:

$$EE\% = (\text{total peptide} - \text{free peptide in supernatant})/\text{total peptide} \times 100\%$$

$$LC = (\text{quantified amount of peptide loaded into GPs})/(\text{weight of GPs})$$

**Determination of semi-permeability**. GPs solution (5 mg/mL) was pretreated with Rhodamine for 10 min to get Rhodamine@GPs, and then the mixture was centrifuged at 8000 rpm/min for 10 min. Removing the supernatant, the precipitation was resuspension in 1 × PBS, followed by repeating this operation for extra four times. Leica LSM 800 laser scanning confocal microscope and

fluorescence spectrophotometer (HITACHI F-7000) were used to detect the fluorescent intensity changes of Rhodamine in GPs during the centrifugations. The fluorescent intensity changes of supernatants were also detected by fluorescence spectrophotometer and calculate the encapsulation and releasing efficiency.

Similarly, the detection of the fluorescent intensity changes of TPE in TPE-Q18H@GPs solution and the fluorescent intensity changes of Rhodamine in TPE-Q18H@GPs solution with Rhodamine during the centrifugations were also performed as the same operation.

**Determination of catalytic kinetics**. The catalytic activities of TPE-Q18H nanofibers and TPE-Q18H@GPs were measured using 4-nitrophenyl acetate (pNPA) as the substrate. The hydrolytic product of pNPA was 4-nitrophenol (pNP) with an absorption peak at 400 nm. pNPA stock solution (160 mM) was dissolved in acetonitrile and kept at $-20\,°C$. All enzymatic assays were initiated by adding $15\,\mu L$ of pNPA with a series of concentrations (20–160 mM) to $285\,\mu L$ of TPE-Q18H solution ($33.42\,\mu M$) or TPE-Q18H@GPs solution (TPE-Q18H: $33.42\,\mu M$; GPs: 1 mg/mL). After 1 min, the reaction mixture was fast centrifuged (5000 rpm/min for 5 s), and the supernatant was separated to detect the absorbance of pNP at 400 nm by UV–vis spectrophotometer (HITACHI-U3900). Each reported result was the average of three measurements. The calculations of initial catalytic rate also subtracted the self-hydrolysis rate of pNPA in $1 \times$ PBS to ensure that the activities were purely from TPE-Q18H or TPE-Q18H@GPs. The extinction coefficient of 4-nitrophenol was according to the previous report[65]. The kinetic constants were calculated by the Michaelis–Menten equation with the non-linear fitting method.

**Protective effect of GPs**. The solution of TPE-Q18H nanofibers or TPE-Q18H@GPs were preheated at 50 °C and 70 °C for 30 min and then cooled down to room temperature. Measure the catalytic activities of TPE-Q18H and TPE-Q18H@GPs after heating and compare the catalytic rate with the unheating group (37 °C). In this study, the blank is the self-hydrolysis rate of pNPA in $1 \times$ PBS, the calculations of initial catalytic rate subtracted the self-hydrolysis rate of pNPA in $1 \times$ PBS to ensure that the activities were purely from TPE-Q18H or TPE-Q18H@GPs.

The solution of TPE-Q18H nanofibers or TPE-Q18H@GPs were incubated with trypsin (0.1 mg/mL) for 5 min and then the catalytic rate was measured, compare the catalytic rate with the untreated group. In the trypsin untreated group, the blank is self-hydrolysis rate of pNPA in $1 \times$ PBS, the calculations of initial catalytic rate subtracted the self-hydrolysis rate of pNPA in $1 \times$ PBS to ensure that the activities were purely from TPE-Q18H or TPE-Q18H@GPs. In the trypsin treated group, for the trypsin could also hydrolyze pNPA, the calculations of initial catalytic rate of TPE-Q18H with trypsin subtracted the hydrolysis rate of pNPA in trypsin to ensure that the activities were purely from TPE-Q18H. The calculations of initial catalytic rate of TPE-Q18H@GPs with trypsin, subtracted the hydrolysis rate of pNPA in trypsin with GPs to ensure that the activities were purely from TPE-Q18H@GPs.

## Data availability

The authors declare that all data supporting the findings of this study are available within the Article and its Supplementary Information. Supplementary methods and data are available in the online version of the paper. The raw data generated in this study are available from the corresponding author upon reasonable request.

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

## Acknowledgements

The authors acknowledge the financial support from the National Key Research and Development Program of China (2020YFA0907300).

## Author contributions

Y.W. and C.Z. conceived the whole work and designed experiments; D.K. and C.Z. provided funding acquisition; Y.W. and T.P. involved and finished all experiments; X.W. contributed to peptide synthesis, fluorescence spectroscopy, and enzymatic activity measurement; F.S. contributed to peptide synthesis and atomic force microscope measurements; A.L. and Y.T. contributed to transmission electron microscope measurements; T.W. and Q.Z. contributed to data analysis; Y.W., T.P., and C.Z. finished all data analysis and the manuscript; All authors discussed and commented on the manuscript.

## Competing interests

The authors declare no competing interests.
