## [Peer review file · Communications Chemistry]

Reviewers' comments:

Reviewer #1 (Remarks to the Author):

Authors was packaging catalytic peptides in glucan cage particles (GPs) to enhance catalytic activity plus endurance and stability. This reviewer has general questions about this approach. Of course, the protection will be beneficial but the conformation is probably more important. When peptides were assembled in GPs, how do authors know that the peptide assemblies are in the best conformation? Is the confinement really paid off for merely 2-fold enhancement, which I am guessing that it is still not practical as compared to natural enzymes. This issue needs to be addressed otherwise this manuscript could be regarded as a low impact work.

In general, this manuscript is sloppy. There are many typos and mis-labeling figures. (e.g., Fig 2-f in page 8 is really Fig. 2-h). The illustration in Fig. 1 is misleading. There is no evidence that peptides were assembled in the fiber form inside GP (I see no structural difference between Fig. 2-e and 2-f) and pore size of GPs is unknown and not investigated (although a dye molecule seems to be in and out). Nanofibers are in the size range of > 1µm, which can really fit in the 3 µm diameter of GP? Why such as large fibers are not visible in TEM images? How many % of the incubated peptides were taken by GP? What is the uptake kinetics of Rhodamine and releasing kinetics? Protease was used to examine the protective effect via peptide degradation but can protease penetrate inside GPs? Pores are that large on GPs?

Reviewer #2 (Remarks to the Author):

This paper describes the entrapment of a self-assembled peptide with hydrolytic catalytic activity in the pores of glucan microparticles. The system design is based on previous literature about related peptides with a pendant histidine unit that becomes catalytic active, hydrolase-like, upon self-assembly into nanofibers.

The use of glucan provides some advantages to this system:

- a) The self-assembled catalyst is supported into an inert matrix, and consequently, its separation from the reaction media by centrifugation is possible
- b) Notably, the nanofiber catalysts are protected from degradation by temperature or trypsin upon incorporation in the glucan particles.
- c) The catalytic performance of the entrapped nanofibers is somewhat improved.

Overall, the research shows novelty and provides interesting results. However, some essential experimental details are missing, and some issues should be clarified before publication.

- 1) The term "Spatio-temporal control" is used in the title and the main text. Clarify the relevance of this issue compared to other cases where such control is not achieved.
- 2) The text states, "was dissolved in solution (acetonitrile/water=3:7) to get a dispersive solution". What is a dispersive solution? Are the authors talking about dispersion? Is it a suspension or a colloid?
- 3) The encapsulation efficiency is claimed to be over 90%. This result seems doubtful. The loading is performed using a "dispersion" of the peptide in acetonitrile/water, and the loading efficiency is determined after centrifugation by UV-Vis study of the supernatant. However, the centrifugation probably also removes "dispersed"/precipitated peptides outside the glucan particles. Therefore, a control experiment is required in the absence of glucan.
- 4) Calculated errors should be provided for the kinetic constants.
- 5) How can the thermal stability induced by entrapment into glucan be explained?
- 6) The hydrolysis rate of PNPA in the studied medium without the peptide should be indicated numerically
- 7) In the experimental section: "The lyophilized powder was resuspended with 1× PBS to obtain TPE-Q18H@GPs" What volume of PBS?
- 8) "The UV-vis spectra of TPE-Q18H in different concentrations were obtained on a UV-vis spectrophotometer..." What solvent was used?

9) The kinetic measurements seem to be flawed by this ambiguous description: "After 1 minute, the reaction mixture was fast centrifuged" Are the authors reporting kinetic data for a reaction time of one minute that requires centrifugation before measurement of absorbance?! What is "fast" centrifugation? In situ measurements of the hydrolysis product seem mandatory in this case or the use of longer reaction times.

Reviewer #3 (Remarks to the Author):

The paper reports spatiotemporally confined encapsulation of self-assembled fibers into hollow glucan particles to construct enzymatic compartments. The resulting biomimetic system shows improved catalytic efficiency and stability against perturbation conditions. The results are well addressed and could be interesting to the broad audience. Overall, this is an interesting concept, and it can be practically useful. This manuscript offers a brand-new perspective inspired by the natural structure of cell compartments, and I recommend acceptance of this manuscript after the following minor issues are addressed.

1. The authors claimed that one advantage of such strategy was that the artificial catalysts could be retained within the cavities of the GPs, while the free biomacromolecules are able to enter or leave the cavities freely, and they used rhodamine dye as an indicator to show this semi-permeability. Yet, the biomacromolecule model was not disclosed in the manuscript.
2. From the sequence design of TPE-Q18H, the KRKR sequence was chosen to increase the solubility of TPE moiety, while, as I know, in another published work the acidic amino acid residues were ever used to make the TPE-hybrid peptide soluble in water. So, did the authors have other considerations about the sequence design to choose such an alternative way.
3. The authors monitored the catalytic activities in several different conditions individually. The blanks were used for these conditions should be referred in detail, separately.
4. The scale bars in Figure 2 should be labelled more clearly, especially for the AFM image.

Response to reviewers

Reviewer 1

Q1. When peptides were assembled in GPs, how do authors know that the peptide assemblies are in the best conformation?

Response: Thanks a lot for reviewer's comments. The optical method to directly observe the morphologies of peptide assemblies inside GPs is limited due to the existence of GPs shell. In this regard, the conformation of TPE-Q18H in GPs was first characterized by incubating with thioflavin T (Th T), a typical β -sheet indicator. When the self-assembly of the peptide was triggered by salt, both TPE-Q18H nanofibers and TPE-Q18H@GPs exhibited pronounced enhancement at 488 nm maxima of Th T fluorescent spectrum (Figure 2h in our manuscript), providing strong evidence of the formation of β -sheet conformation.

In our revised work, we supplemented the Fourier Transform infrared spectrometer (FTIR) to further prove the conformation of TPE-Q18H in GPs. The FTIR spectra of assembled TPE-Q18H, empty GPs, and TPE-Q18H @GPs were obtained by the KBr tablet method. Absorbance spectrum of assembled TPE-Q18H and TPE-Q18H @GPs shared two peaks in the amide I region at 1631 cm^{-1} and 1674 cm^{-1} (Figure R1), indicating the existence of antiparallel β -sheet structures^{1,2} which was in agreement with Th T results. The antiparallel β -sheet conformation meant that there were multiple interactions among β -sheet strands, indicating the formation of nanofibrillar structures. This data was also placed in the revised supporting information as Supplementary Figure 7.

Figure R1: FTIR spectra of assembled TPE-Q18H, empty GPs, and TPE-Q18H @GPs.

Based on the results above, we believe that the peptides assembled inside GPs and the structural features of these assemblies were consistent with those of TPE-Q18H in the GP-free state.

Q2. Is the confinement really paid off for merely 2-fold enhancement, which I am guessing that it is still not practical as compared to natural enzymes?

Response: Thank you very much for your concern about our design. It is still the challenge for most studies of artificial enzymes that the catalytic efficiencies of artificial enzymes are in different order of magnitudes compared to those of natural enzymes. While, here we just wanted to claim a new strategy of constructing bioinspired compartments by immobilizing artificial enzymes into natural porous materials, to improve the activities and especially the stability against perturbation conditions with

easier recycling, which avoided the extensive consumption and waste of natural products.

Moreover, based on current strategy, our ongoing work is proposing an approach to fabricate protocells by *in situ* self-assembly of multiple natural enzymes to carry out cascade reactions within GPs. To explore the universality of our strategy, we have immobilized a modified green fluorescence protein with a designed peptide sequence capable of self-assembly (EGFP-peptide) in GPs. This model EGFP-peptide can self-assembly inside GPs like the way of TPE-Q18H. Confocal laser scanning microscopy (CLSM) was used to observe the green fluorescence of EGFP-peptide inside GPs (Figure R2a), indicating that protein could be retained within GPs by salt-triggered peptide assembly. Almost no fluorescence was recorded in the supernatant solution and the encapsulation efficiency is 95.51% in the fluorescence measurement (Figure R2b). It lays a foundation for the subsequent research to immobilize natural enzymes into GPs. Hence, we believe the robust and recyclable properties of our work would pay off, and the spatiotemporally confined *in situ* self-assembly strategy has significant potential in construction of unnatural organelles.

Figure R2. The fabrication of EGFP-peptide@GPs protocells. (a) CLSM image of EGFP-peptide@GPs. Scale bar: 7.5 μm . (b) Fluorescence spectra of supernatant of EGFP-peptide@GPs and free EGFP-peptide. $\lambda_{\text{ex}}=405$ nm.

Q3. There are many typos and mis-labeling figures. (e.g., Fig 2f in page 8 is really Fig. 2h).

Response: We really appreciate reviewer's comments and the typos and mislabeling mistakes have been revised, which has been highlighted in our revised manuscript.

Q4. The pore size of GPs is unknown and not investigated (although a dye molecule seems to be in and out).

Response: Thank you for your valuable suggestion. In our submission, we could check the pore size of GPs under SEM with high resolution in Figure 2c, and furthermore, in our revised manuscript we supplementary measured the surface properties of GPs by nitrogen adsorption isotherms (Figure R3). The adsorption-desorption curve of GPs is a type II isotherm, and the BET surface area and average pore diameter of GPs were 12.84 m^2/g and 4.14 nm respectively, indicating that not only small molecules but also biomacromolecules can pass through the pore easily. This result has been placed and highlighted in our revised manuscript as shown in Figure 2d.

Figure R3. Nitrogen adsorption isotherms and pore size distribution (inset) calculated based on the DFT method for GPs.

Q5. The illustration in Fig. 1 is misleading. There is no evidence that peptides were assembled in the fiber form inside GP (I see no structural difference between Fig. 2-e and 2-f). Nanofibers are in the size range of $> 1 \mu\text{m}$, which can really fit in the $3 \mu\text{m}$ diameter of GP? Why such as large fibers are not visible in TEM images?

Response: As we mentioned in the response to **Q1**, the assembly of peptides in GPs was comprehensively characterized by ThT assay and FTIR. Hence the illustration of Fig. 1 is based on solid conclusions in the revised manuscript.

About the size issue, we claimed that TPE-Q18H was not at an assembled state when peptide entered into GPs, since we did not add any salt into the system to trigger the assembly process at this time point. After regulating ionic strength, TPE-Q18H self-assembled into soft and flexible nanofibers in the cavity of GPs *in situ*. Please check our description of the preparation of TPE-Q18H@GPs in the “exploration of enzymatic compartments fabrication strategy” part in our revised manuscript.

The shell of GPs composed of branched β -(1,3; 1,6)-glucan has certain thickness, and the pore size is rather small compared to the length of peptide nanofibrils, so it is very hard to visualize the morphology of nanofibers in the GPs. However, TPE-Q18H@GPs showed some corrugations compared with GPs in TEM (Figure 2e and Figure 2f) when zooming the images. In addition, the existence of assembled TPE-Q18H inside GPs has been proved by multiple methods such as Th T, FTIR and the activity assay.

Q6. How many % of the incubated peptides were taken by GP?

Response: Thank you for your comments. As shown in Supplementary Figure 5, we established the standard curve of TPE-Q18H, measured the absorbance of TPE-Q18H supernatant at 254 nm (Supplementary Figure 6), and then the encapsulation efficiency was calculated to be 98.23%. Mirza, Z. et al. reported $>95\%$ protein package inside the GPs using the same preparation method.³

Q7. What is the uptake kinetics of Rhodamine and releasing kinetics?

Response: Thanks for your valuable suggestion. We supplement the uptake and releasing kinetics of Rhodamine in GPs (Supplementary Figure 9 in our revised supporting information). Rhodamine and empty GPs incubated for 10 minutes. One group was filtered and measured leachate's fluorescence to

measure the uptake. Another group was centrifuged and resuspended 5 times to measure the releasing of Rhodamine from GPs. Free Rhodamine solution was prepared for calculations.

Calculate the encapsulation and releasing efficiency using the formulas below:

Uptake efficiency= (FL intensity of free Rhodamine - FL intensity of leachate) / fluorescence of free Rhodamine \times 100% (FL means fluorescent intensity)

Releasing efficiency= FL intensity of supernatant / (FL intensity of free Rhodamine -FL intensity of leachate) \times 100%

In particular, the calculation of first centrifugation releasing is the FL intensity of supernatant of the first centrifugation minus the FL intensity of the leachate.

As shown in Figure R4, 50% Rhodamine was released from GPs in the first centrifugation, and all Rhodamine was released after 5 centrifugations.

Figure R4. The releasing kinetics of Rhodamine from GPs during centrifugations.

Q8. *Protease was used to examine the protective effect via peptide degradation but can protease penetrate inside GPs? Pores are that large on GPs?*

Response: Thank you for your significant reminder. The pore average size was characterized to be around 4.14 nm, so theoretically Trypsin (maximum diameter 4.2 nm) could enter into GPs. To investigate whether the protease Trypsin (24 kDa) can penetrate GPs and TPE-Q18H@GPs, we supplement experiments using a green fluorescent protein (PSP2, 27 kDa) as a model, because trypsin and PSP2 share similar sizes. PSP2 was incubated with GPs or Q13H@GPs (Q13H replaces TPE-Q18H to eliminate the effect of TPE on fluorescence of PSP2 protein) solution for 10 min, and CLSM was used to detect the fluorescent intensity of PSP2 in GPs and Q13H@GPs. The green fluorescence of PSP2 was observed in GPs (Figure R5a) and Q13H@GPs (Figure R5b). Moreover, the average intensity in Q13H@GPs was lower than that in GPs because peptide Q13H nanofibers occupied some space (Figure R5c). With this result, we can conclude that Trypsin can penetrate into GPs, and we speculate that the limited internal space of GP makes the conformation of peptide assemblies relatively fixed, so the structural stability was improved.

Figure R5. Protein permeability of GPs and Q13H@GPs was assessed by CLSM. (a) the CLSM image of GPs incubated with PSP2. Scale bar: 7.5 μm . $\lambda_{\text{ex}}=405$ nm. (b) the CLSM image of Q13H@GPs incubated with PSP2. Scale bar: 7.5 μm . $\lambda_{\text{ex}}=405$ nm. (c) The average intensity of the fluorescence region in CLSM calculated by Image J software.

Reviewer 2

Q1. The term “Spatio-temporal control” is used in the title and the main text. Clarify the relevance of this issue compared to other cases where such control is not achieved.

Response: Spatio-temporal control means controlling the assembly occurs when and where. In this research, we fabricated the enzymatic compartment TPE-Q18H@GPs in a spatiotemporally controllable manner. The time point of peptide self-assembly is triggered by salt. After non-assembled TPE-Q18H was encapsulated into GPs, 1×PBS buffer was added to trigger the self-assembly of TPE-Q18H in the cavity of GPs.

Compartments without spatio-temporal control cases are like the construction of enzyme@MOF capsules. Enzyme molecules were embedded into the MOF support matrix, in which the MOF crystals are synthesized in the presence of enzymes^{4,5}. In this way, enzymes were randomly immobilized into MOF without any control on timing and position.

Q2. The text states, “was dissolved in solution (acetonitrile/water=3:7) to get a dispersive solution”. What is a dispersive solution? Are the authors talking about dispersion? Is it a suspension or a colloid?

Response: Thank you for pointing out the inaccurate statement in our manuscript. Peptide TPE-Q18H dissolved very well in solution (acetonitrile/water=3:7), so it is actually a clear solution and the peptide was in the monomer state. The expression “was dissolved in solution (acetonitrile/water=3:7) to get a dispersive solution” has been changed into “was dissolved in solution (acetonitrile/water=3:7) to get a clear solution and the peptide was in the monomer state” in the revised manuscript to avoid this misunderstanding.

Q3. The encapsulation efficiency is claimed to be over 90%. This result seems doubtful. The loading is performed using a “dispersion” of the peptide in acetonitrile/water, and the loading efficiency is determined after centrifugation by UV-Vis study of the supernatant. However, the centrifugation probably also removes “dispersed”/precipitated peptides outside the glucan particles. Therefore, a control experiment is required in the absence of glucan.

Response: As corrected in the **Q2**, peptide TPE-Q18H was dissolved very well in solution (acetonitrile/water=3:7) and the peptide monomers entered GPs by capillarity owing to the porous

structure of GPs. The assembly of TPE-Q18H occurred *in situ* in the cavity of GPs by the addition of 1×PBS buffer and the assembled nanofibers within GPs could not escape from GPs because of their micrometer length size, even during centrifugation, while the monomers and the nanofibers out of GPs would be separated into the supernatant

Q4. *Calculated errors should be provided for the kinetic constants.*

Response: Thanks for your comments. The calculated errors have been added to Figure 5d and the data has been updated in the description part in our revised manuscript.

Esterase kinetic constants		
	TPE-Q18H	TPE-Q18H@GPs
K_m (mM)	2.91±0.45	0.50±0.22
V_{max} ($\mu\text{M min}^{-1}$)	36.65±2.27	14.89±1.12
$k_{cat} \times 10^{-3}$ (s^{-1})	18.28±1.13	7.43±0.56
k_{cat}/K_m ($\text{M}^{-1} \text{s}^{-1}$)	6.29±1.05	14.82±6.61

Figure R6. Updated esterase kinetic constants of TPE-Q18H nanofibers and TPE-Q18H@GPs.

Deviations of k_{cat} and K_m were the standard errors of fitting and the deviation of k_{cat}/K_m was calculated by the equation:

$$\sigma_{k_{cat}/K_m} = k_{cat}/K_m \times \sqrt{(\sigma_{k_{cat}}/k_{cat})^2 + (\sigma_{K_m}/K_m)^2}$$

Q5. *How can the thermal stability induced by entrapment into glucan be explained?*

Response: Thanks a lot for your concern. Normally, no matter natural enzymes or artificial enzymes lose activities at high temperature mainly due to the change of their best conformation. The limited internal space of GPs made the conformation of peptide assemblies relatively fixed, so the thermal stability was improved.

Q6. *The hydrolysis rate of pNPA in the studied medium without the peptide should be indicated*

Response: The self-hydrolysis rate of pNPA (1 mM) was 14.79 $\mu\text{M}/\text{min}$ in 1× PBS. Meanwhile, in the calculations of the initial catalytic rate of TPE-Q18H or TPE-Q18H@GPs, the self-hydrolysis of pNPA was also subtracted to ensure that the activities were purely from TPE-Q18H or TPE-Q18H@GPs.

Q7. *In the experimental section: “The lyophilized powder was resuspended with 1× PBS to obtain TPE-Q18H@GPs” What volume of PBS?*

Response: The volume of 1× PBS was 1 mL. In the experimental section, we have changed the description “The lyophilized powder was resuspended with 1 mL 1× PBS to obtain a well-dispersed TPE-Q18H@GPs particles solution and stocked at 4 °C. When measuring the enzyme activities, add another 4 mL 1× PBS to get a diluted solution (TPE-Q18H: 33.42 μM ; GPs: 1 mg/mL)”.

Q8. *“The UV-vis spectra of TPE-Q18H in different concentrations were obtained on a UV-vis spectrophotometer... “What solvent was used?”*

Response: The solvent was 1× PBS. In the experimental section, we have changed the description into “The UV-vis spectra of TPE-Q18H 1× PBS solution in different concentrations was obtained on a UV-vis spectrophotometer from 190 nm to 450 nm to make a standard curve”.

Q9. *The kinetic measurements seem to be flawed by this ambiguous description: “After 1 minute, the reaction mixture was fast centrifuged” Are the authors reporting kinetic data for a reaction time of one minute that requires centrifugation before measurement of absorbance? What is “fast” centrifugation? In situ measurements of the hydrolysis product seem mandatory in this case or the use of longer reaction times.*

Response: We are appreciative of your suggestion. we did not measure the hydrolysis product *in situ* because the 3-5 μm diameter of GPs will interfere with absorbance, leading to large errors. On the other hand, centrifugation allowed the products to escape from GPs. The fast centrifugation is 5000 rpm using Mini Centrifuge (SCIOLOGEX S1010E) for 5 seconds. The ambiguous description has been revised in the manuscript.

Reviewer 3

Q1. *The authors claimed that one advantage of such strategy was that the artificial catalysts could be retained within the cavities of the GPs, while the free biomacromolecules are able to enter or leave the cavities freely, and they used rhodamine dye as an indicator to show this semi-permeability. Yet, the biomacromolecule model was not disclosed in the manuscript.*

Response: We appreciate your valuable suggestion. We supplement the macromolecule model using a green fluorescent protein (PSP2 27 kDa). PSP2-loaded GPs (PSP2@GPs) were prepared by simply pretreating empty GPs with PSP2. After centrifuging and resuspending three times, CLSM was used to detect the fluorescent intensity changes of PSP2 during this process. Before the centrifugation, the solid green fluorescence of PSP2 was observed in the interior of GPs (Figure R7a), indicating that PSP2 could enter GPs freely. After three times centrifugation, the remaining fluorescence of PSP2 in GPs decreased (Figure R7b), showing PSP2 could release from GPs. Moreover, kinds of literature reported the encapsulation of macromolecules, such as protein,³ DNA,⁶ and RNA,⁷ and they adopted to form nanoparticles to prevent the escape of macromolecules.

Figure R7. Permeability to macromolecule of GPs was assessed by CLSM. (a) the CLSM image of GPs incubated with PSP2 and resuspensions after three times centrifugation. Scale bar: 7.5 μm.

$\lambda_{\text{ex}}=405$ nm. (b) The average intensity of the fluorescence region in CLSM image calculated by Image J software.

Q2. *From the sequence design of TPE-Q18H, the KRKR sequence was chosen to increase the solubility of TPE moiety, while, as I know, in another published work the acidic amino acid residues were ever used to make the TPE-hybrid peptide soluble in water. So, did the authors have other considerations about the sequence design to choose such an alternative way.*

Response: We have ever synthesized the TPE-Q18H with the EDED sequence instead of KRKR sequence, but its solubility was lower than the KRKR sequence. Furthermore, GPs are negatively charged glycan particles with lots of hydroxyl groups, so we hypothesized that TPE-Q18H with KRKR sequence would be easier to interact with and enter into GPs. To load more peptides into GPs, we chose the designed TPE-Q18H with KRKR sequence.

Q3. *The authors monitored the catalytic activities in several different conditions individually. The blanks were used for these conditions should be referred to in detail, separately.*

Response: Thanks for your significant reminder. In the parts of determining catalytic kinetics and thermostability, the blank was the self-hydrolysis rate of pNPA in 1× PBS, and the calculations of initial catalytic rate subtracted the self-hydrolysis rate of pNPA in 1× PBS to ensure that the activities were purely from TPE-Q18H or TPE-Q18H@GPs.

In the part of protecting from protease digestion, the blank was the self-hydrolysis rate of pNPA in 1×PBS in trypsin untreated group, and the calculations of the initial catalytic rate were described as above. In the trypsin treated groups, for the trypsin could also hydrolyze pNPA, the calculations of the initial catalytic rate of TPE-Q18H with trypsin needed to subtract the hydrolysis rate of pNPA by trypsin to ensure that the activities were purely from TPE-Q18H. The calculations of the initial catalytic rate of TPE-Q18H@GPs with trypsin subtracted the hydrolysis rate of pNPA in trypsin with GPs to ensure that the activities were purely from TPE-Q18H@GPs. We also added the description in detail in the revised manuscript.

Q4. *The scale bars in Figure 2 should be labeled more clearly, especially for the AFM image.*

Response: We thank the reviewer for the careful review and kind suggestion. We have clearly labeled the scale bars in Figure 2, checked and modified the other bars in the revised manuscript and supporting information.

References

- 1 Hamley, I. W. *et al.* Self-assembly of a modified amyloid peptide fragment: pH-responsiveness and nematic phase formation. *Macromol. Biosci.* 10, 40-48, doi:10.1002/mabi.200900217 (2010).
- 2 Reches, M. & Gazit, E. Designed aromatic homo-dipeptides: formation of ordered nanostructures and potential nanotechnological applications. *Phys. Biol.* 3, S10-19, doi:10.1088/1478-3975/3/1/S02 (2006).
- 3 Mirza, Z., Soto, E. R., Dikengil, F., Levitz, S. M. & Ostroff, G. R. Beta-Glucan Particles as Vaccine Adjuvant Carriers. *Methods Mol. Biol.* 1625, 143-157, doi:10.1007/978-1-4939-7104-6_11 (2017).
- 4 Liang, K. *et al.* Biomimetic mineralization of metal-organic frameworks as protective coatings for biomacromolecules. *Nat. Commun.* 6, 7240, doi:10.1038/ncomms8240 (2015).

- 5 Shieh, F. K. et al. Imparting Functionality to Biocatalysts via Embedding Enzymes into Nanoporous Materials by a de Novo Approach: Size-Selective Sheltering of Catalase in Metal-Organic Framework Microcrystals. *J. Am. Chem. Soc.* 137, 4276-4279 (2015).
- 6 Soto, E. R. & Ostroff, G. R. Characterization of multilayered nanoparticles encapsulated in yeast cell wall particles for DNA delivery. *Bioconj. Chem.* 19, 840-848, doi:10.1021/bc700329p (2008).
- 7 Aouadi, M. et al. Orally delivered siRNA targeting macrophage Map4k4 suppresses systemic inflammation. *Nature* 458, 1180-1184, doi:10.1038/nature07774 (2009).

REVIEWERS' COMMENTS:

Reviewer #1 (Remarks to the Author):

There is significant improvement on this manuscript with more data sets. However, there are still a few questions needed to be addressed with new data.

1. Now pore size is identified and release kinetics was also studied and it is very fast. Then, it comes a question that how long the peptide assembly takes and what time point it shows enzymatic activity. These numbers should be compared to justify all of authors' hypothesis.

2. Are Figure R2 and its explanation in revised manuscript? It should be in to support a tremendous claim like these peptide-incorporating GPs are potentially developed as unnatural organelles. Ideally, authors should incorporate another catalytic peptide in GP and simultaneously assemble both and analyze how they can respectively trigger two distinct enzymatic reactions. There are a plenty of papers that used other cage-like polymer or protein shells to incorporate multiple enzymes and thus authors' data could be compared and commented with respect to them.

Reviewer #2 (Remarks to the Author):

The remarks raised in my first review have been properly answered in this revised version.

Reviewer #3 (Remarks to the Author):

The authors addressed all issues we concerned. I recommend to publish.

Response to Reviewers #1

Q1. Now pore size is identified and release kinetics was also studied and it is very fast. Then, it comes a question that how long the peptide assembly takes and what time point it shows enzymatic activity. These numbers should be compared to justify all of authors' hypothesis?

Response: Thank you very much for your concern about our design. To explore how long the peptide assembly takes and what time point it shows enzymatic activity, we first monitored the fluorescence intensity change of Th T that represents the conformation change of peptide TPE-Q18H, and then measured the catalytic activities of TPE-Q18H and TPE-Q18H@GPs before and after adding salt respectively. As shown in Figure RR1 a, for the free peptide TPE-Q18H, the fluorescence intensity of Th T increased immediately after adding salt and tended to be stable after $T=30$ min, indicating that the assembly reached equilibrium at this time point. In comparison, the fluorescence intensity of Th T reached the highest level immediately after triggering the assembly in TPE-Q18H@GPs (Figure RR1 b), demonstrating that peptide TPE-Q18H assembled completely in GPs within one minute. From these observations, we hypothesize that the ultra-fast assembly process of TPE-Q18H@GPs is because the peptides are already highly concentrated in GPs and the very close distance accelerates the process of nucleation and recruitment of free peptide molecules. Furthermore, the activities of TPE-Q18H and TPE-Q18H@GPs were measured along with the assembly process. The hydrolysis rates of pNPA increased rapidly in both TPE-Q18H (Figure RR1 c) and TPE-Q18H@GPs (Figure RR1 d) groups after adding salt, demonstrating that the catalytic performance emerged as long as the assembly started. These results and discussion have been highlighted and put into the revised manuscript and supporting information.

Figure RR1. The fluorescence and UV-Vis investigation of the assembly and activity kinetics of TPE-Q18H and TPE-Q18H@GPs upon salt. (a) The fluorescence intensity of Th T incubated with TPE-Q18H; (b) The fluorescence intensity of Th T incubated with TPE-Q18H@GPs. (c) Catalytic curves of pNPA (10 mM) hydrolysis in the presence of TPE-Q18H. (d) Catalytic curves of pNPA (10 mM) hydrolysis in the presence of TPE-Q18H@GPs. $[Th\ T]=250\ \mu M$.

Q2. Are Figure R2 and its explanation in revised manuscript? It should be in to support a tremendous claim like these peptide-incorporating GPs are potentially developed as unnatural organelles. Ideally, authors should incorporate another catalytic peptide in GP and simultaneously assemble both and analyze how they can respectively trigger two distinct enzymatic reactions. There are a plenty of papers that used other cage-like polymer or protein shells to incorporate multiple enzymes and thus authors' date could be compared and commented with respect to them?

Response: Thanks for your comment. We have followed your suggestion about incorporating another catalytic peptide in GPs, and here we designed and constructed bifunctional enzymatic compartments with hydrolase and peroxidase-like activities by immobilizing hemin, the key part of the peroxidase catalytic center, onto the surface of TPE-Q18H nanofibers with a peptide/hemin ratio of 10/1 (Figure RR2 a). The hemin bound nanofibers (TPE-Q18H/Hemin₁₀) were characterized using UV-Vis spectroscopy (Figure RR2 b). The free hemin chloride displayed a Soret peak at 384 nm along with a shoulder at 365 nm, indicating the presence of dimeric (μ -oxo bihemin) along with some monomeric hemin hydroxide (haematin).¹ In contrast, hemin bound nanofibers (TPE-Q18H/Hemin₁₀) displayed a broad Soret band at 400 nm, similar to hemin in aqueous micelle solutions and proteins, suggesting monomeric hemin chloride.^{2,3} The absorption peak at 330 nm was from TPE. We then explored the peroxidase-like activity of hemin using 3,3',5,5'-tetramethylbenzidine (TMB) and H₂O₂ as the substrates. Peroxidases facilitated the oxidation of a colorless TMB to a blue product with maximum absorbance at 652 nm in the presence of H₂O₂. The activity was characterized by monitoring the absorbance at 652 nm of reaction supernatants. As shown in Figure RR2 c, TPE-Q18H showed no catalytic activity but TPE-Q18H/Hemin₁₀ exhibited a higher activity than that of free hemin. Similarly, GPs and TPE-Q18H@GPs were proved to have no contribution to the catalytic activity, while TPE-Q18H/Hemin₁₀@GPs carried out the oxidation redox reaction. The absorbance peak of 652 nm was not detected in Hemin@GPs, which may be due to too few products and not completely escaping from the GPs.

In addition, the hydrolytic activities were measured using pNPA as the substrate (Figure RR2 d). GPs, Hemin, Hemin@GPs did not show hydrolytic activity. TPE-Q18H, TPE-Q18H@GPs and TPE-Q18H/Hemin₁₀@GPs exhibited dramatic hydrolytic activity towards pNPA, which was in consistent with previous results.

In summary, we have successfully constructed a dual artificial enzyme system in porous GPs, which exhibits both hydrolase and peroxidase-like activities. As suggested by editor, we prefer to put Figure R2, Figure RR2 and the related discussion into the transparent peer review scheme.

Figure RR2. The construction and activity measurement of bifunctional enzymatic compartments with hydrolase and peroxidase-like activities. (a) Illustration of enzymatic compartments with hydrolase and peroxidase-like activities. (b) UV-vis spectra of free hemin and TPE-Q18H/Hemin₁₀. (c) UV-vis spectra of reaction supernatants after reaction for 5 min in the presence of GPs, Hemin, Hemin@GPs, TPE-Q18H, TPE-Q18H/Hemin₁₀, TPE-Q18H@GPs and TPE-Q18H/Hemin₁₀@GPs. (d) Catalytic curves for hydrolysis of pNPA (4 mM) in the presence of GPs, Hemin, Hemin@GPs, TPE-Q18H, TPE-Q18H/Hemin₁₀, TPE-Q18H@GPs and TPE-Q18H/Hemin₁₀@GPs.

References

- 1 Silver, J. & Lukas, B. Mössbauer studies on protoporphyrin IX iron(III) solutions. *Inorg. Chim. Acta* **78**, 219-224, doi:10.1016/s0020-1693(00)86516-0 (1983).
- 2 Monzani, E. *et al.* Enzymatic properties of human hemalbumin. *Biochim. Biophys. Acta* **1547**, 302-312, doi:10.1016/s0167-4838(01)00192-3 (2001).
- 3 Shantha, P. K., Saini, G. S. S., Thanga, H. H. & Verma, A. L. Photoreduction of iron protoporphyrin IX chloride in non-ionic triton X-100 micelle studied by electronic absorption and resonance Raman spectroscopy. *J. Raman Spectrosc.* **32**, 159-165, doi:10.1002/jrs.674 (2001).